# Effect of Hot-Wet Storage Aging on Mechanical Response of a Woven Thermoplastic Composite

**Theofanis S. Plagianakos** [1,†,*] ⬤**, Kirsa Muñoz** [2,†]**, Diego Saenz-Castillo** [3,†]**,
Maria Mora Mendias** [3,†]**, Miguel Jiménez** [2,†] **and Vasileios Prentzias** [1,†]

1   Hellenic Aerospace Industry S.A., GR 32009 Schimatari, Greece; prentzias.vasilios@haicorp.com
2   Element Materials Technology Seville, C/Wilbur y Orville Wright 1, Aerópolis,
    41300 San José de la Rinconada, Seville, Spain; kirsa.munoz@element.com (K.M.);
    miguel.jimenez@element.com (M.J.)
3   FIDAMC, Foundation for the Research, Development and Application of Composite Materials,
    Avda Rita Levi Montalcini 29, Tecnogetafe, 28906 Getafe, Madrid, Spain; Diego.Saenz@fidamc.es (D.S.-C.);
    maria.mora.mendias@fidamc.es (M.M.M.)
*   Correspondence: plagianakos.theofanis@haicorp.com; Tel.: +30-226-205-2247
†   These authors contributed equally to this work.

**Abstract:** The effect of hot-wet storage aging on the mechanical response of a carbon fiber polyether ether ketone (PEEK)-matrix woven composite has been studied. A wide range of static loads and selected cyclic load tests on the interlaminar fatigue strength were performed. Static tests were conducted in batch mode, including on- and off-axis tension, compression, flexure, interlaminar shear strength (ILSS) and fracture tests in Modes I, II and I/II. Respective mechanical properties have been determined, indicating a degrading effect of aging on strength-related properties. The measured response in general, as well as the variance quantified by batch-mode test execution, indicated the appropriateness of the applied standards on the material under consideration, especially in the case of fracture tests. The material properties presented in the current work may provide a useful basis towards preliminary design with PEEK-based woven thermoplastic composites during service in aerospace applications.

**Keywords:** carbon fibre thermoplastic composite; PEEK matrix; woven; aging; mechanical testing; static and fatigue

## 1. Introduction

Continuous-fiber reinforced thermoplastic composites are gaining attention in the aerospace industry for exhibiting advantages compared to thermoset composites, such as design and manufacturing flexibility, including multiple post-forming processes, and capability of being processed by a large range of traditional machining methods, fast cycle time and recyclability. As far as their mechanical performance is concerned, their enhanced impact resistance is a very attractive feature for selecting them in demanding lightweight applications. Moreover, in woven ply configurations they yield less anisotropic mechanical properties, which could be desirable in the context of conceptual design. In this context, the experimental determination of their mechanical properties over a wide range of static and fatigue mechanical tests, as well as quantification of the effect of aging on these properties, are extremely important for design and in-service monitoring purposes.

The main advantages of thermoplastic composites in comparison with thermoset composites have been very well described in the scientific literature to date. Excellent mechanical properties, good behaviour against impact, no need of cold storage owing to their long shelf-life and no chemical

reaction during consolidation, which opens the possibility of short-time processing are among their most appealing features [1–4]. There is a significant amount of literature focusing on mechanical characterization of continuous-fiber thermoplastic composites. Chu et al. [5] characterized 3-D braided Graphite/polyether ether ketone (PEEK) composites by static tensile tests and experimentally determined time-dependent mechanical properties at various temperatures by performing relaxation experiments and dynamic mechanical analysis (DMA) tests. Hufenbach et al. [6] studied the strain-rate dependency of the mechanical properties of three thermoplastic composite materials, including impact energy absorption, at different temperatures, and additionally studied the effect of fiber–matrix interphase modification on transverse tensile strength. Liu et al. [7] developed a damage model accounting for fiber failure and matrix cracking and validated it with an open-hole compression test on woven PEEK specimens. Kuo et al. [8] performed 4-point bending tests on thermoplastic composites and studied the effect of molding temperature on flexure properties and failure modes. Boccardi et al. [9] used infrared tomography to study the effect of frequency on the temperature of cantilever glass- and jute-based woven thermoplastic composite specimens subjected to cyclic bending. Sorentino et al. [10] fabricated and characterized polyethylene naphthalate (PEN) thermoplastic composites at 100 °C by 3-point bending, DMA and impact tests. As far as shear loading characterization is concerned, Hufenbach et al. [11] performed Iosipescu tests on woven thermoplastic composites and used high-speed camera and digital image correlation for studying the deformation and failure in order to develop material modeling strategies in commercial finite element software. Zenasni and coworkers [12,13] experimentally studied the effect of fiber material and weave pattern on the fracture response of polyetherimide (PEI)-matrix-based woven thermoplastic composite specimens in Mode I and Mode II.

　　　The effect of aging on static and dynamic response of engineering thermoplastics has been studied since the early 1990s. As a general conclusion, PEEK polymers (and, therefore, PEEK-reinforced composites) are not the thermoplastic polymers which may absorb the highest level of water. It has been reported by Buchman and Isayev [14] that PEEK composites may absorb approximately 0.2% of humidity, in comparison with other thermoplastic polymers, which may absorb above 1%. This is mainly caused by the semi-crystalline structure of PEEK, in comparison with other thermoplastic polymers which have a larger amorphous part. Béland [15] concluded that semicrystalline thermoplastics/carbon composites may absorb also around 0.2% of humidity while carbon/amorphous thermoplastic may absorb beyond 0.8%. The effect of thermal aging on carbon-fibre reinforced PEEK has been studied by Buggy and Carew [16], who performed static and fatigue flexure tests. Aging has been applied by two different methods: by storage for up to 76 weeks at high temperatures and by thermal cycling from room temperature (RT) to 250°C. They indicated a low sensitivity of the laminate flexural modulus and strength to ageing at 120 °C. Kim et al. [17] developed an analytical model for predicting the flexural properties degradation at high temperatures (540–640 °C) and performed 4-point bending tests for validation purposes. In the context of an implant application study, Schambron et al. [18] experimentally determined the effect of environmental ageing on static and cyclic flexure response of carbon-fibre/PEEK coupons and reported superior fatigue resistance compared to stainless steel. An experimental comparison of the bearing strength of woven-ply-reinforced thermoplastic or thermoset laminates at 120 °C after hygrothermal aging has been performed by Vieille et al. [19].

　　　Thus, it can be concluded that the effect of aging on the mechanical response of woven thermoplastic composites has yet not been quantified over a wide range of mechanical tests. The present work aims to close this void by presenting an extensive test campaign performed to assess the mechanical properties of a high-performance woven carbon-fiber reinforced thermoplastic material in non-aged and aged conditions. Material characterization has been achieved by conducting mechanical tests according to (static tests) or based on (fatigue tests) ASTM and ISO standards, such as tension, compression, in-plane and interlaminar shear, flexure, Mode I, II and I/II fracture. Properties derived from static tests are reported in batch mode, to provide a measure of the test-type non-linearity and testing procedure repeatability. Moreover, the effect of aging is assessed by measuring mechanical properties

after specimen environmental conditioning in hot-wet storage. Finally, albeit the main objective of the current work is to provide experimental data, in order to check the applicability of linear elastic fracture mechanics based on effective material properties, interlaminar fracture test cases in Mode I and II have been modelled in a commercial finite element software and the predictions were compared with measured values. The main conclusion of the present work is that aging leads to significant degradation of the strength in engineering woven thermoplastic composites, while stiffness-related properties seem to be rather insensitive to aging.

## 2. Materials and Methods

### 2.1. Specimen Manufacturing

Thermoplastic composite coupons for mechanical properties' characterization were cut from a set of carbon fiber/PEEK plates manufactured by FIDAMC (Getafe, Spain). The engineering polymer selected for this study consisted of Tenax®TPCL PEEK-4-40-HTA40 3K supplied by Toho-Tenax (Tokyo, Japan). The material was carbon-fiber-reinforced PEEK (CF/PEEK) fabric 5HS (5 harness satin), 0.31 mm nominal thickness per ply, 40% of resin weight fraction and a fiber areal weight (FAW) of 285 gsm. The stacking sequence of the thermoplastic laminates varied depending on the mechanical testing and defined by the standards, explained in detail in the next sub-sections. It should be noted that, due to the fact that the used CF/PEEK fabric was a 5HS (non-symmetrical), the 0° direction of each ply was defined as the warp direction of the roll material, as suggested by the material supplier.

On the basis of previous experience with other candidate manufacturing methods [20], the selected manufacturing process for the consolidation of the laminates was compression molding by hot-platen press. The utilized equipment is located at the FIDAMC facilities (Getafe, Spain). Each laminate was first-hand laid-up with the help of a manual welder and then located in a metallic frame which acted as material retainer. Two polyimide sheets with a release agent were placed at both sides of the laminate, acting as release films. Two metallic caul plates were also used in order to obtain a proper flat-surface finish. The consolidation cycle consisted of a heating ramp at approximately 2 °C/min up to a consolidation dwell of 30 min at 400 ± 10 °C with an applied pressure of 1 MPa.

### 2.2. Hot-Wet Storage Aging

During its service life, an aircraft is exposed to high temperatures and high levels of humidity. The properties of composite materials may be affected as a consequence of moisture absorption and high temperatures. A faithful replication of the environmental exposure during aircraft operation would require a cyclic conditioning procedure between hot/humid and cold/dry conditions, as dictated by relevant standards (MIL-STD-810 or other). In the proposed work, we have followed an accelerated conditioning procedure on the basis of common practice [21], which would form a small part of an extended experimental aging campaign towards material airworthiness certification.

In order to evaluate the degradation of mechanical properties, an environmentally conditioned testing scenario was considered according to ASTM D5229/D5229M. The selected method is a recommended pre-test conditioning method, consistent with the recommendations of the Composite Materials Handbook-17 [22]. Specifically, the procedure that has been followed in these test series is a conditioning procedure, BHEP, which covers non-ambient moisture conditioning of material coupons from the same batch as the ones tested, widely known as traveler specimens, in a humidity chamber (-H-) at a prescribed constant, conditioning the environment to equilibrium (-E-), periodic (-P-) coupon weighing being required. Two different types of test series were covered. On the one hand, room temperature (RT) tests were conducted on specimens without any previous conditioning. Room Temperature conditions are controlled to meet standard laboratory atmosphere conditions, which according to ASTM D5229/D5229M, are 23 ± 2 °C and 50% ± 10% relative humidity. On the other hand, parallel test series were performed on specimens that have been previously conditioned. These are known as Hot Wet (HW) tests.

The conditioning parameters are usually fixed according to the conditions to which aircraft structure may be subjected during its service life. This commonly means an equilibrium moisture weight in an 85% relative humidity environment and a temperature of 70 °C. Nevertheless, in order to achieve a lower completion time for the testing campaign, accelerated conditioning was carried out. In order to achieve a faster aging process, conditioning was performed at a higher temperature (80 ± 3 °C), under the same relative humidity (85% ± 5%). It was checked that glass transition temperatures for the assessed material, as provided by the manufacturer, were significantly higher than the accelerated conditioning temperature. By proceeding this way, a decrease in conditioning periods was achieved, speeding the rate of testing.

## 2.3. Experimental Methods

All mechanical tests presented in this work have been performed according to (static tests) or based on (fatigue tests) relevant ASTM or ISO standards. Each standard has been reported in the title of relevant the subsection of the following section. Static tests have been performed in a five specimen batch mode to provide a statistically valid representation of a material sample response under static loads. For the sake of cost savings in the case of such an extended test campaign, three specimens per test type have been tested in fatigue, each one at a load level representing a fatigue cycle regime: low cycle-($1.0 \times 10^4$ cycles), medium cycle-($1.0 \times 10^5$ cycles) and high cycle-fatigue ($1.0 \times 10^6$ cycles). Each test was assumed to have concluded with a failure if either the specimen failed due to rupture or if a load-displacement slope decrease of at least 10% was detected compared to the slope at the 100[th] loading cycle. A stress ratio of $R = \sigma_{min}/\sigma_{max} = 0.1$ has been considered, which is typical for the characterization of carbon-fiber reinforced plastics (CFRP) under dynamic loading [23,24]. Regarding testing frequency, high frequencies may produce an increase in the specimen temperature, which leads to mechanical properties degradation [25]. To avoid this, a frequency of 5 Hz was applied in all fatigue tests and the temperature of the specimens was monitored by means of a thermocouple placed on each. The laboratories involved in the campaign considered a range of specimen temperatures from RT up to 35 °C to be acceptable. For the selected frequency of testing, no specimen temperature reached 35 °C at any of the tests performed, and thus no effect of frequency on the results was assumed. According to the testing strategy, the fatigue endurance limit was set to $1.0 \times 10^6$ cycles.

All tests of the non-aged composite system (RT) have been conducted in Hellenic Aerospace Industry (HAI) facilities on an INSTRON 8801 (Norwood, MA, USA) hydraulic testing machine at room temperature ($T = 25 \pm 1$ °C, 45% ± 5% humidity), equipped by an INSTRON load cell with a range up to 100 kN. In the case of fracture tests, a 5 kN Omega load cell has been used. In the case of double-cantilever beam (DCB) and mixed-mode bending (MMB) tests, a Philips SPC2050NC digital camera and Debut v4.08 by NCH software (Greenwood Village, CO, USA—non-commercial use edition) have been used to record crack propagation, whereas, in end-notch flexure (ENF) tests, crack propagation has been visually monitored. The camera-to-specimen distance was ca. 130 mm and focus has automatically been applied at a 640 × 80 pixel resolution and a rate of 30 frames per second.

HW tests have been performed on different Universal Testing Machines in Element facilities. Static tension (0°, ±45°, 90°) and compression (0°, 90°) tests were performed on a Zwick Z100 BS1, (Kennesaw, GA, USA) equipped with a Zwick load cell up to 100 kN. Static flexure, interlaminar shear strength (ILSS) and interlaminar fracture toughness tests were performed on a Zwick Retroline (INSTRON 5866) Universal Testing Machine, equipped with an INSTRON load cell having a range up to 10 kN. All static HW tests have been conducted under controlled temperature (70 °C) using thermocouples inside a temperature chamber (Thermcraft). Regarding fatigue, HW tests, tension (0°, ±45° laminates) and ILSS tests were performed on a Universal Dynamic Testing Machine INSTRON 8872, equipped with an INSTRON load cell up to 25 kN. For fatigue tension (90° laminate), a Universal Dynamic Testing Machine INSTRON 8801, equipped with an INSTRON load cell up to 100 kN, was used. In all fracture tests crack propagation has been visually monitored.

HW fatigue tests were performed at RT right after being environmentally conditioned. That choice was based on the assumption that there is an irreversible degradation of the material due to specimen conditioning. As reported in the literature [26], long-term environmental conditioning leads to irreversible changes that cause permanent property alterations within the matrix, the fiber surfaces and the fiber/matrix interface. Therefore, by testing unaged and aged specimens under fatigue loading at RT conditions, it is possible to evaluate the effect of permanent degradation caused by hygrothermal conditioning.

### 2.4. Finite Element Models

The numerical simulations for pure fracture Modes I and II have been performed using MSC MARC finite element (FE) software [27]. In both cases, cohesive elements have been integrated into the models in order to evaluate the prediction of delamination propagation with experimental results. The composite thermoplastic material is modelled using property values at ply level extracted from mechanical tests. The constitutive relation of the cohesive elements is based on Linear Elastic Fracture Mechanics (LEFM) and an exponential damage law is used [28]. The cohesive properties of the interface have been derived by experimentally determined values for normal traction (strength—Section 3.3.1), interlaminar shear traction (strength—Section 3.5.1) and critical energy release rates in Mode I (Section 3.5.3) and Mode II (Section 3.5.4). From these values, normal traction and fracture toughness in Mode I have been adapted according to the strategy formulated in [29] in order to yield a critical opening displacement allowing for solution convergence. As discussed in Sections 3.5.3 and 3.5.4, the modeling approach adopted herein, which has been successfully applied in unidirectional thermoset materials [28], failed to accurately estimate the maximum load in both fracture modes of the woven thermoplastic material.

In order to simplify the analysis of both DCB and ENF tests, 2D models have been created using plane strain full-integration elements (Type 11) for the bulk material and cohesive elements (Type 186) for the interface. There are four elements through the thickness for the DCB model and five elements for the ENF model because of the difference in specimen thickness. An element length of 0.75 mm has been selected for composite and interface in order to achieve an acceptable convergence rate.

## 3. Results and Discussion

The test campaign for material characterization in terms of mechanical properties includes static and fatigue tests. Static tests have been performed for the determination of basic mechanical properties to feed the structural design phase of thermoplastic parts, as well as for quantification of the degradation of these properties due to aging during service. Fatigue tests provide an overview of fatigue endurance of the material via measured S–N curves and validate the trend observed in the static tests concerning the effect of aging. Specifically, static tests include tension, compression, in-plane shear, flexure, interlaminar shear, and interlaminar fracture toughness in Mode I, II and I/II. Fatigue tests include interlaminar shear strength. The specimen nominal dimensions in fatigue were identical to the ones used in the corresponding static case. Wherever available, the measured data are compared with values provided by the manufacturer [30].

### 3.1. Specimen Dimensions and Lamination

The specimen dimensions have been selected in order to comply with the relevant standard. The lamination has been determined on the basis of relevant standards, while available experimental capabilities have also been taken into account, in order to remain within the measurable load range of available equipment. In Table 1, the geometry and lamination of the specimens are presented. The test type description is provided as T-0 (tension warp), T-90 (tension weft), C-0 (compression warp), C-90 (compression weft), FLEX (flexure), T-45 (tension ±45), ILSS (interlaminar shear strength), ILFT I (interlaminar fracture toughness Mode I), ILFT II (interlaminar fracture toughness Mode II), ILFT I/II (interlaminar fracture toughness Mixed-Mode).

**Table 1.** Specimens dimensions and lamination ($ denotes odd number of plies, as: FLEX—13 plies, ILSS—19 plies).

| Specimen Data | T-0 | T-90 | C-0 | C-90 | FLEX | T-45 | ILSS | ILFT I | ILFT II | ILFT I/II |
|---|---|---|---|---|---|---|---|---|---|---|
| Lamination | $[(0/90)]_3$ | $[(0/90)]_6$ | $[(0/90)]_{3S}$ | $[(0/90)]_{3S}$ | $[(0/90)_7]_\$$ | $[(45/-45)_8]_S$ | $[(0/90)_{10}]_\$$ | $[(0/90)]_{5S}$ | $[(0/90)]_{7S}$ | $[(0/90)]_{7S}$ |
| Length (mm) | 250 | 175 | 79.4 | 79.4 | 154 | 250 | 40 | 125 | 160 | 155 |
| Width (mm) | 15 | 25 | 12.7 | 12.7 | 13 | 25 | 12 | 20 | 25 | 25 |
| Thickness (mm) | 0.9 | 1.9 | 1.9 | 1.9 | 4 | 4.8 | 6 | 3 | 4.2 | 4.2 |

## 3.2. Specimen Aging

For each test type, the respective moisture absorption curve, measured from corresponding traveler specimens (in-plane dimensions $25 \times 25$ mm), is reported. Figure 1 provides an overview of the conditioning process. On the basis of these curves, the moisture content uptake rate and water content at saturation during conditioning have been extracted in Table 2. For the determination of the slope, a measurement at saturation and the initial measurement have been used.

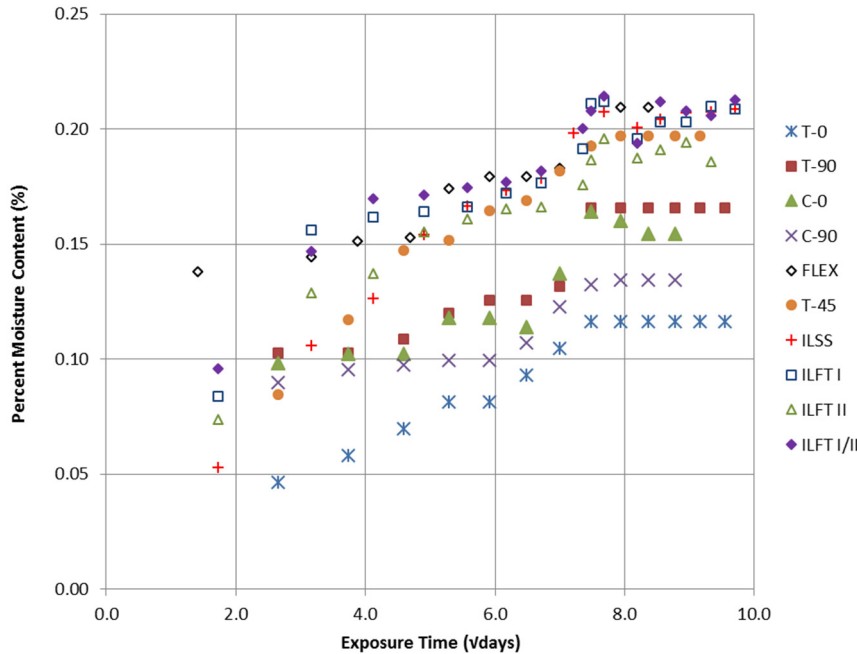

**Figure 1.** Conditioning curves for test types considered; the exposure time is expressed in square root of days.

**Table 2.** Moisture uptake parameters extracted from conditioning curves.

| Moisture | T-0 | T-90 | C-0 | C-90 | FLEX | T-45 | ILSS | ILFT I | ILFT II | ILFT I/II |
|---|---|---|---|---|---|---|---|---|---|---|
| Uptake Rate (%/days$^{0.5}$) | 0.013 | 0.013 | 0.014 | 0.009 | 0.011 | 0.021 | 0.026 | 0.018 | 0.021 | 0.017 |
| Maximum (%) | 0.116 | 0.166 | 0.164 | 0.134 | 0.209 | 0.197 | 0.209 | 0.212 | 0.196 | 0.214 |

In order to comment on the trends observed in the conditioning process, the thickness of the relevant traveler specimen should be taken into account, since in-plane dimensions have been identical for all traveler specimens. It may be concluded that thicker traveler specimens, such as those considered for in-plane and interlaminar shear tests (including fracture toughness) yield generally higher moisture uptake rates and maximum values. This trend is attributed to the fact that moisture penetrates more easily across the sides of the specimens along thickness, which are uncoated compared to the upper and lower faces. Moreover, it is highly related to the effect of aging on the response in each test, as will be explained in the following sections.

Concerning comparison with data reported in the open literature, the maximum value of moisture uptake obtained for T-0 is very close to data reported by other researchers [31] (0.13%) and within the range set by previous studies on the moisture absorption of engineering thermoplastics [14,15].

### 3.3. Determination of Fiber-Dominated Properties

3.3.1. Static Tension of Woven Laminate Warp Direction (ASTM D3039)

The head velocity was 2 mm/min, as dictated by the standard. Strains were measured by strain gages: biaxial rosettes have been attached on each specimen, whereas an additional longitudinal strain gage has been attached on the opposite side of one specimen to ensure that bending remained below 10% throughout the test. Data for stress vs. longitudinal strain for RT and HW specimens are presented in Figure 2. It should be noted that the maximum load may differ from the value shown, as strain gages occasionally got detached at a lower load level. Moreover, in the legend of Figure 2 the custom naming convention pattern followed in all subsequent data figures is provided.

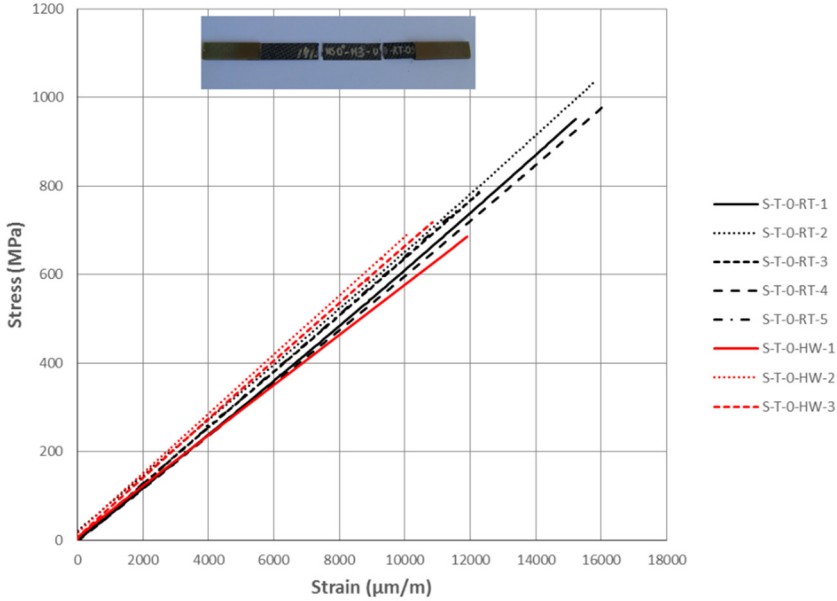

**Figure 2.** Static tension of warp-directional laminate: stress vs strain data and characteristic failure mode at RT. The description of tests follows a custom naming convention: S—static, T—tension, 0—warp direction, RT—room temperature test, HW—hot wet test, last digit—specimen number.

Elaboration of the above static tension test data yields the mechanical properties shown in Table 3. Properties are provided in terms of mean value (MV) and coefficient of variance (CV). CV is directly related to the standard deviation (STDEV) as: CV=STDEV/MV, and thus STDEV could be alternatively used for reporting the deviation intervals in the experimentally determined values.

The strain range between 1000 and 3000 $\mu\varepsilon$ has been used for extraction of the tensile elastic modulus. The respective values provided in the manufacturer datasheet according to ISO 527-4 Type 3 are also listed for comparison.

**Table 3.** Effect of hot-wet storage aging on mechanical properties derived from static tension tests on warp-directional laminates.

| Property | RT | | | HW | |
|---|---|---|---|---|---|
| | Mean Value | CV (%) | Manufacturer Datasheet | Mean Value | CV (%) |
| Tensile Strength (MPa) | 889 | 13 | 955 | 701 | 2 |
| Tensile Modulus (Gpa) | 62.7 | 4 | 60 | 63.8 | 9 |
| Poisson ratio | 0.078 | 29 | - | 0.025 | 6 |

It may be observed that the elastic modulus is insensitive to aging (2% difference), whereas strength is strongly affected in a negative sense (21% reduction). This trend agrees well with results obtained by Solvay for PEEK thermoplastic polymer prepregs [32], and may be at least partially explained by the fact that the modulus is evaluated in the linear strain range (1000–3000 μm/m), while strength involves larger strains. It is reasonable to assume that in such a high demanding situation in terms of strain, the load-bearing capacity of the material appears to be sensitive to aging. As far as the Poisson ratio is concerned, it also seems to be affected, however, its measure is small and thus not expected to be of major importance as a design variable.

A typical failure mode is also illustrated in Figure 2. Lateral failure modes at multiple areas and various locations (LMV according to ASTM D3039) have been primarily observed. This trend has been expected, since the woven pattern of the fabric in the laminate prevents the occurrence of exploding modes. Similar failure patterns have been observed in RT and HW specimens, indicating that failure mode is rather insensitive to aging. Most failure patterns (8 out of 10) included failure near the tabs, indicating the redistribution of stresses along the specimen near/at failure load. The subsequent failure mode is related, partially at least, to inertia forces following the first failure, to a 3-D stress state involving local stress fields near the tabs, and to free-edge effects. The present approach focuses on the determination of the effect of hot-wet storage aging on the global mechanical properties of the woven thermoplastic material, while a detailed study of the failure mechanisms involved [11] would require special equipment, which has not been available in the context of this work.

### 3.3.2. Static Tension of Woven Laminate Weft Direction (ASTM D3039)

The selection of a different lamination compared to the warp-directional specimens, as listed in Table 1, is explained by the fact that the tests shown in this work are part of a larger test campaign where unidirectional laminates of various material systems are mechanically evaluated. Wherever possible, identical specimen dimensions for all material systems have been selected with the purpose of maximizing comparability for the same mechanical property evaluation. In this context, the tests on the weft-directional specimens have been similar to the warp-directional ones in terms of head velocity and strain gauge placement.

Stress vs. longitudinal strain data for RT and HW specimens are presented in Figure 3.

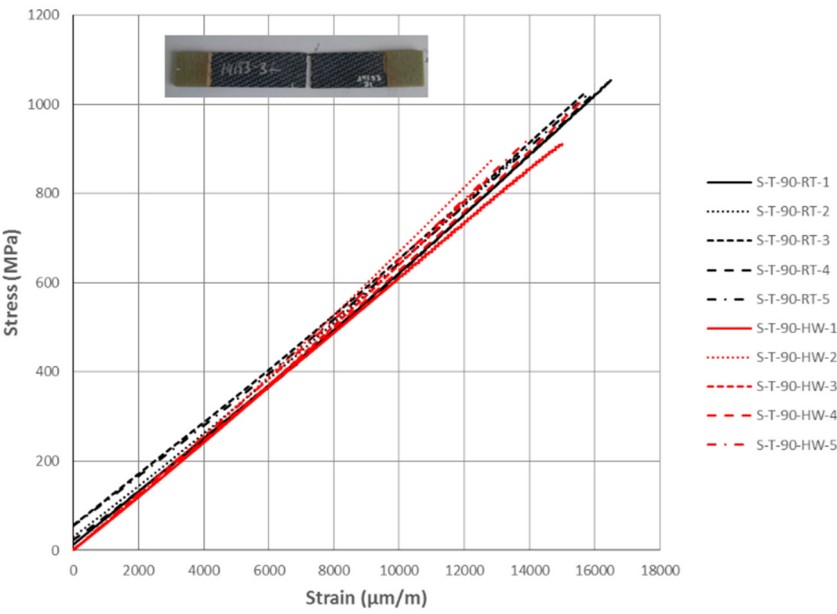

**Figure 3.** Static tension for weft-directional laminate: stress–strain data and characteristic failure mode.

Elaboration of the above static tension test data yields the mechanical properties shown in Table 4. The same analysis was applied as in the case of the warp-directional static tension tests. The respective values provided in the manufacturer datasheet according to ISO 527-4 type 3 are also listed for comparison.

**Table 4.** Effect of hot-wet storage aging on mechanical properties derived from static tension tests on weft-directional laminates.

| Property | RT | | | HW | |
|---|---|---|---|---|---|
| | Mean Value | CV (%) | Manufacturer Datasheet | Mean Value | CV (%) |
| Tensile Strength (MPa) | 973 | 6 | 909 | 907 | 8 |
| Tensile Modulus (GPa) | 56 | 2 | 60 | 61 | 2 |
| Poisson ratio | 0.039 | 16 | - | 0.039 | - |

Concerning the effect of aging, a similar trend is observed, as in the case of the warp-directional specimens. The elastic modulus appears to be slightly (9%) increased due to aging, while strength is slightly (7%) reduced. Such a trend for the modulus has been expected, as both laminations are similar, whereas the reduction in strength is less than expected.

A typical failure mode of an RT specimen is also illustrated in Figure 3. Lateral failure modes have been observed in all specimens, being less distributed along the specimen compared to the case of warp-directional laminates. Most specimens (7/10) failed in the vicinity of the tabs. These failures are attributed to pre-stress, due to excessive pressure on the tabs at the grips of the machine in order to avoid slippage during loading.

### 3.3.3. Static Compression of Woven Laminate Warp Direction (ASTM D695)

Two batches in total, each consisting of 5 RT and 4 HW specimens, were tested. The head velocity was 1.3 mm/min. Strains were measured by two unidirectional strain gages, each attached on the narrow part of each side of the dogbone specimen to ensure that bending remained below 10% up to buckling.

A typical load vs. longitudinal strain measurement is presented in Figure 4. A typical failure mode is also shown. All specimens failed under intralaminar shear.

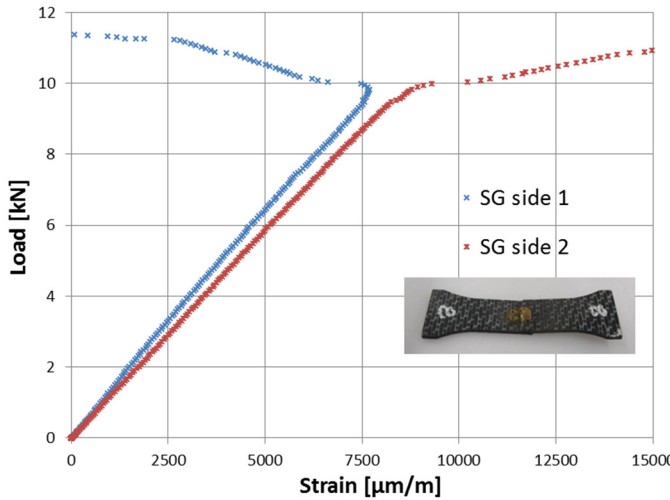

**Figure 4.** Compression test for warp-directional laminate: typical measured load vs longitudinal strain data and typical failure mode (SG: strain gauge).

Elaboration of the above static compression test data in the strain range between 1000 and 3000 με yields the elastic modulus shown in Table 5. The respective values provided in the manufacturer datasheet according to EN 2850 are also listed for comparison.

**Table 5.** Effect of hot-wet storage aging on compressive properties of warp-directional laminates.

| Property | RT | | | HW | |
| --- | --- | --- | --- | --- | --- |
| | Mean Value (ASTM D695) | CV (%) | Manufacturer Datasheet (EN 2850) | Mean Value (ASTM D695) | CV (%) |
| Compressive Modulus (GPa) | 55.0 | 3 | 59.2 | 60.3 | 2 |
| Compressive Strength (MPa) | 536 | 7 | 725 | 514 | 4 |

It may be observed that compressive modulus and strength exhibit low sensitivity (less than 10%) to aging. This trend agrees with the results reported in [32] for a similar material. The values measured for the modulus are close to those provided by the manufacturer, as they do not depend on the specimen geometry accounted for in the two standards. On the contrary, the higher thickness in most of the specimens due to the consideration of tabs in EN 2850 leads to higher strength values compared to the ones measured in the current study.

### 3.3.4. Static Compression of Woven Laminate Weft Direction (ASTM D695)

Two batches in total, each consisting of 5 RT and 4 HW specimens, were tested. The head velocity was the same as in the warp-directional case and strains were also measured accordingly. A typical load vs. longitudinal strain data curve is presented in Figure 5 along with a typical failure mode. All specimens failed under intralaminar shear.

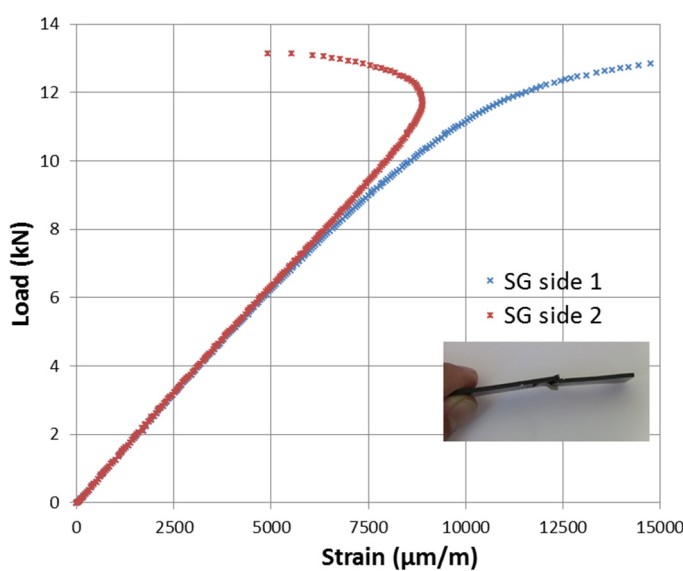

**Figure 5.** Compression test for weft-directional laminate: typical measured load vs. longitudinal strain data and typical failure mode.

Elaboration of the above static compression test data in the strain range between 1000 and 3000 $\mu\varepsilon$ yields the elastic modulus shown in Table 6.

**Table 6.** Effect of hot-wet storage aging on compressive properties of [90] laminates.

| Property | RT | | | HW | |
| --- | --- | --- | --- | --- | --- |
| | Mean Value (ASTM D695) | CV (%) | Manufacturer Datasheet (EN 2850) | Mean Value (ASTM D695) | CV (%) |
| Compressive Modulus (GPa) | 54.4 | 2 | 57.9 | 58.2 | 1 |
| Compressive Strength (MPa) | 541 | 4 | 712 | 504 | 9 |

As expected, the trends observed in the weft-directional lamination are similar to those reported in the warp-directional one (Table 5). Compressive modulus and strength exhibit low sensitivity (less

than 10%) to hot-wet aging. The moduli measured are close to the value provided by the manufacturer, whereas there is a significant deviation in strength due to the application of different standards encompassing different specimen geometries.

### 3.3.5. Static Flexure (ASTM D7264)

Four-point bending tests have been performed in order to determine static flexural properties. Support span was 128 mm in order to comply with the span to thickness ratio suggested by the ASTM standard. The deflection was measured using an extensometer in a configuration similar to an LVDT. The head velocity was 1.5 mm/min.

The load vs. mid-span deflection for RT and HW specimens is presented in Figure 6, along with a characteristic failure mode.

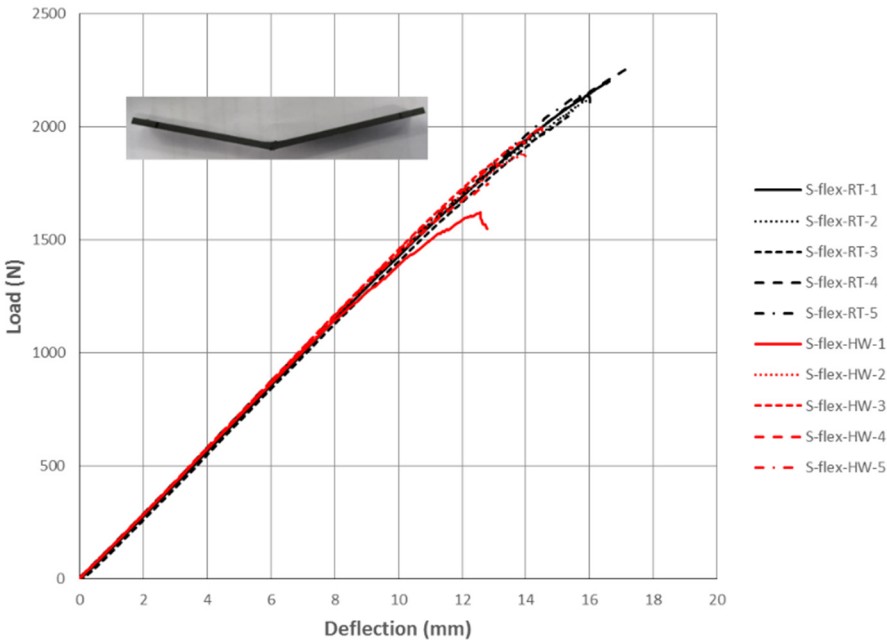

**Figure 6.** Static flexure: load-deflection data and typical failure mode.

Elaboration of the above test data yields the mechanical properties shown in Table 7. Respective values provided for a thinner lamination in the manufacturer datasheet according to EN 2562-A are also listed for comparison.

**Table 7.** Effect of hot-wet storage aging on the mechanical properties derived from static flexure tests.

| Property | RT | | | HW | |
|---|---|---|---|---|---|
| | Mean Value | CV (%) | Manufacturer Datasheet | Mean Value | CV (%) |
| Flexural Strength (MPa) | 974 | 4 | 1225 | 877 | 7 |
| Flexural Modulus (GPa) | 60.1 | 2 | 66.2 | 66.7 | 1 |
| Failure strain (μm/m) | 17337 | 4 | - | 13894 | 6 |

The elastic modulus appears to be slightly increased due to aging (10% increase), as in the case of tensile tests. On the other hand, aging leads to a drop of 16% in flexural strength and 20% in failure strain. Experimental studies reported in the literature [16] showed that thermal aging at 120 °C did not affect flexural modulus and strength. Therefore, the results presented herein indicate that it is the wet storage that mainly leads to strength degradation in a PEEK-matrix based composite configuration.

As far as failure modes of the specimens are concerned, all specimens failed under tension at the bottom surface, either at (5/10, failure description TAB according to ASTM D7264) or between

loading noses (5/10, failure description TBB). According to the standard, these failure modes indicate valid flexural strength. This point is further supported by the acceptable variance in experimentally determined values for strength.

### 3.4. Determination of In-Plane Matrix-Dominated Properties

Static Tension of ±45 Woven Laminate (ASTM D3518)

In order to measure in-plane shear properties of the woven composite material, static tension tests on ±45 laminates have been performed. The head velocity was 2 mm/min, whereas strains were measured in the same manner as the other tensile tests presented above.

The stress vs. shear strain data for RT and HW specimens are presented in Figure 7, respectively. Each strain gage failed at approximately 15,000 $\mu\varepsilon$. The achieved range allowed for the evaluation of the 0.2% offset (yield) strength. Nevertheless, the load continued to rise after the failure of the strain gages and eventually reached a plateau at about 23 kN. Thus, the ultimate strength has been evaluated from the maximum load. At that stage, the plastic deformation of the specimens was visible with the naked eye, albeit no rupture of the specimen has occurred. That trend indicates a ductile response of the material, similar to that of metals, excluding failure in terms of breakage.

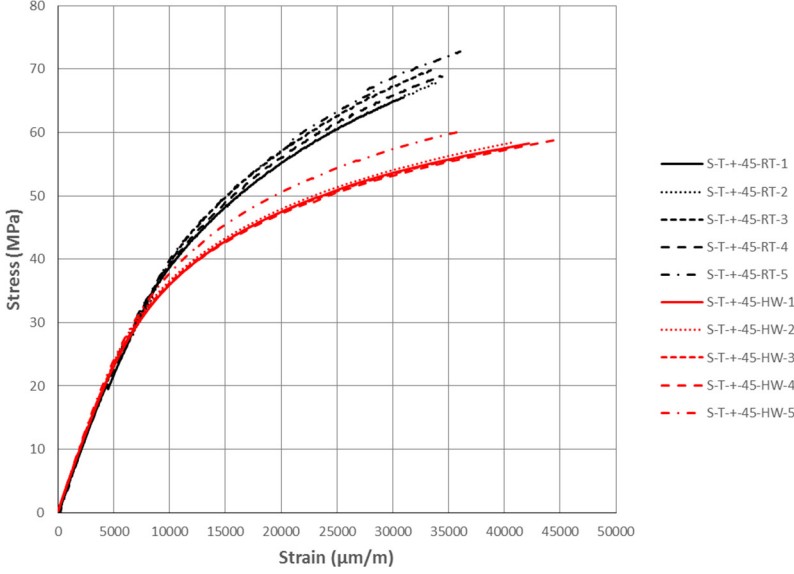

**Figure 7.** Effect of aging on stress–strain curves of in-plane shear strength (IPSS) test.

Elaboration of the above static tension test data yields the mechanical properties shown in Table 8. Respective values provided in the manufacturer datasheet (measured by using a proprietary testing method) are also listed for comparison.

**Table 8.** Effect of hot-wet storage aging on the mechanical properties derived from static tension tests on [±45] laminates.

| Property | RT | | | HW | |
|---|---|---|---|---|---|
| | Mean Value | CV (%) | Manufacturer Datasheet | Mean Value | CV (%) |
| Offset Strength (MPa) | 42.1 | 3 | - | 37.7 | 0.3 |
| Shear Modulus (GPa) | 4.32 | 4 | 5.03 | 4.46 | 2 |
| Maximum Shear Stress (MPa) | 98.4 | 3 | - | 82.9 | 7 |

As was also observed in the behavior of the tensile modulus in the previous tensile cases, the shear modulus, which is evaluated in a range of 2000 to 6000 µm/m, appears to be insignificantly affected

by aging (3% increase). On the other hand, the tangential modulus beyond linear regime, as well as offset strength and maximum stress, are significantly lower in the case of the aged material (10% and 16%, respectively). Thus, the experimental results indicate that aging is more effective in larger strains occuring in the non-linear part of the response, which is prominent in that type of test. Considering that the maximum moisture absorption of in-plane shear specimens is slightly higher than in the tensile specimens, although not much higher, a reasonable phenomenological explanation of this trend is attributed to the thickness to width ratio of the specimens. In the in-plane shear case, the thickness to width ratio is approximately three times higher than in the tensile cases, and thus there is a wider distribution of moisture along the laminate plane, as moisture penetrates across the sides along thickness, which are not protected by a coating layer. As the strains increase, this trend gets more effective on the response and is enhanced by edge effects, which grow dominant. The experimental verification of that phenomenological explanation would be interesting, however, no appropriate equipment was available in the context of the present work to study the micromechanics involved during loading.

*3.5. Determination of Out of Plane Matrix-Dominated Properties*

3.5.1. Static Interlaminar Shear Strength Tests (ILSS) (ASTM D2344)

In the case of ILSS tests the support span was 24 mm, thus yielding a thickness aspect ratio of 4. The deflection was measured using the head displacement of the testing machine. The head velocity was 1 mm/min.

The load vs. mid-span deflection for RT and HW specimens is presented in Figure 8. A typical failure mode is also shown.

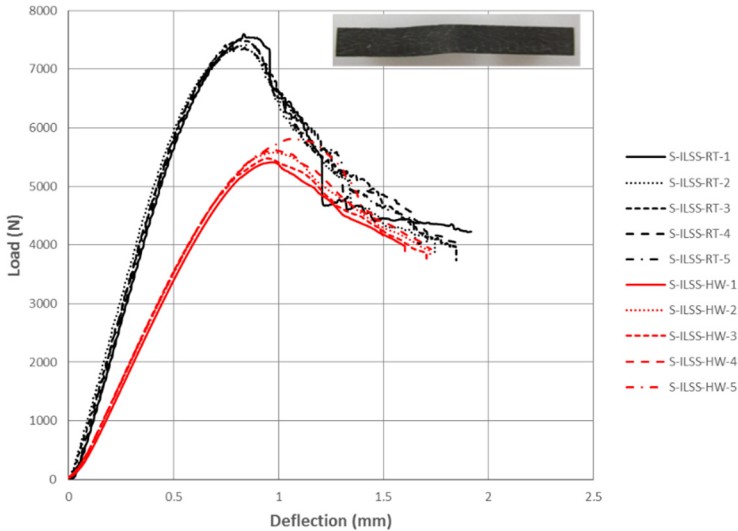

**Figure 8.** Effect of aging on load-deflection curves measured in ILSS test.

Elaboration of the above test data yields the mechanical properties shown in Table 9. For this test type, no data are included in the manufacturer datasheet.

**Table 9.** Effect of hot-wet storage aging on the mechanical properties derived from static ILSS tests.

| Property | RT | | HW | |
|---|---|---|---|---|
| | **Mean Value** | **CV (%)** | **Mean Value** | **CV (%)** |
| Maximum Load (N) | 7453 | 1 | 5581 | 3 |
| Short-beam Strength (MPa) | 84.1 | 1 | 63.6 | 3 |

The interlaminar shear strength appears to be the most sensitive to aging among the properties studied so far, as a reduction of 25% has been experimentally determined. Moreover, the load-displacement slope is also significantly affected within both the linear and non-linear part of the response. This trend has been expected, since the ILSS test is a matrix-dominated test type. In addition, taking into account the dominance of the free-edge effects in such a low thickness aspect ratio (length/thickness) specimens, the ILSS experimental data led credibility to the phenomenological explanation provided in Section 3.4 concerning the effect of thickness to width ratio on the response. The thickness to width ratio of the ILSS specimens is 2.5 and 30 times higher than that of IPSS and tensile specimens, respectively, and thus they are more prone to a wider distribution of moisture in the specimen.

As far as the failure mode is concerned, the specimens failed under inelastic deformation and interlaminar shear. Each of these failure modes are considered acceptable in the standard, however, in the current case the failure appeared to include both modes as a result of the ductility of the matrix compared to thermoset composites.

### 3.5.2. Fatigue Interlaminar Shear Strength (ILSS) Tests

The effect of aging on interlaminar shear fatigue strength is presented in Figure 9. Aging has a clear degrading effect on the fatigue strength of the thermoplastic material, which is obvious in low-, medium- and high-cycle regimes. This trend is in-line with the experimental results of corresponding static tests and supports the argumentation based on the effectiveness of the thickness to width ratio on the enhanced sensitivity of ILSS specimens to hot-wet storage aging. It should be noted that all failures in fatigue have been attributed to a reduction in slope in the load-displacement loop beyond 10%. It should be also indicated that, as these tests are limited in quantity, statistical means cannot be drawn.

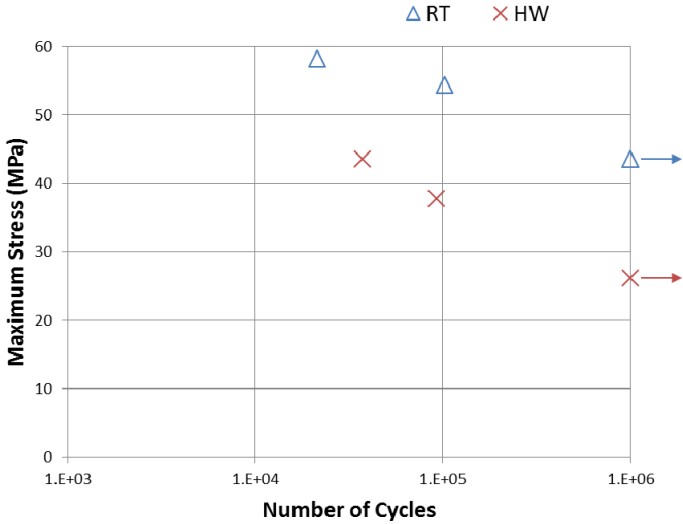

**Figure 9.** Effect of aging on fatigue interlaminar shear strength. Arrows to the right indicate that no failure has occurred.

### 3.5.3. Static Interlaminar Fracture Toughness Tests Mode I (ISO 15024)

According to the standard, an initial delamination of 60 mm has been built in by the intervention of a release film at the middle of the stacking sequence in the fabrication process. A head velocity of 1 mm/min has been applied. An initial loading has been considered for creating a crack propagation of 3–5 mm to eliminate fiber bridging effects, followed by a reloading stage for further crack propagation. In Figure 10a, measured load vs. opening displacement data for RT and HW specimens at the reloading stage are presented. A predicted instance of crack propagation is shown in Figure 10b. An interesting observation during the test was the "stick-slip" response during crack propagation, which resulted

in the usability of few data points for the evaluation of fracture toughness. While in the case of unidirectional (UD) thermoset materials, a stable propagation has been measured [28], it being common practice and, as such, considered in the related standards, in the case of the woven thermoplastic material propagation occurred in maximum of four steps. As shown in Figure 10c, both the upper and lower face have been excessively deformed at each propagation step. Thus, strain energy was stored in the face up to a sudden release. At that point, the crack propagated to an arbitrary distance, indicating a highly non-linear response and leading to a relatively large variance in fracture toughness. This stick-slip response has not been observed in similar tests on PEI matrix-based woven thermoplastics reported in the open literature [13], whereas the values obtained for fracture toughness are in a similar range. Thus, the stick–slip response could be attributed to the type of matrix, adhesion between fibers and matrix and weave type.

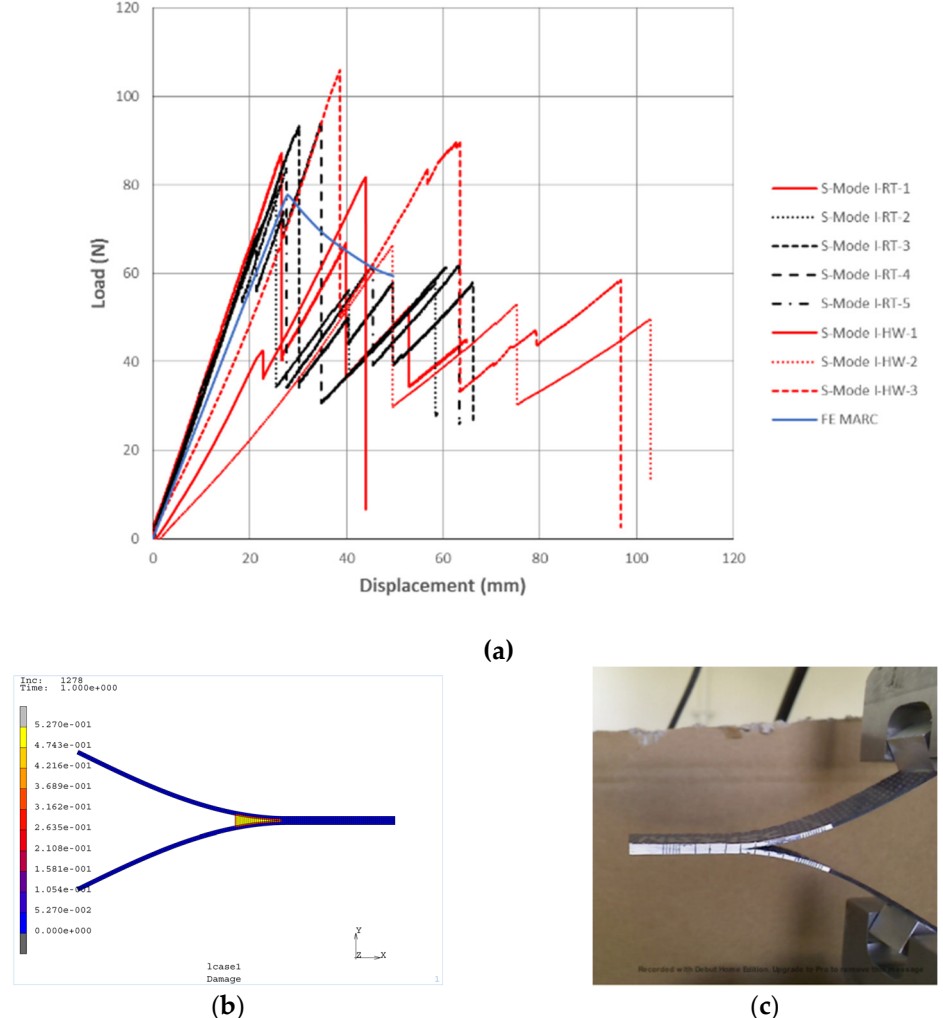

**(a)**

**(b)**    **(c)**

**Figure 10.** Mode I fracture test: (**a**) measured and predicted load vs. opening displacement data; (**b**) instance of crack propagation prediction; (**c**) typical deformed shape during test.

The FE model, which is based on LEFM, succeeds at predicting the slope of the load-displacement curves, whereas deviation is observed for maximum load. The deviation observed indicates that detailed unit-cell models and the application of appropriate failure criteria for woven thermoplastic materials should be applied in order to improve this prediction.

Elaboration of the measured data on the basis of Corrected Beam Theory (accounting for large-displacement correction) yields the interlaminar fracture toughness values presented as mean values in Table 10. For this test type no data are included in the manufacturer datasheet.

**Table 10.** Effect of hot-wet storage aging on interlaminar fracture toughness in Mode I.

| Property | RT | | HW | |
|---|---|---|---|---|
| | Mean Value | CV (%) | Mean Value | CV (%) |
| ILFT Mode I (J/m$^2$) | 2490 | 12 | 3304 | 19 |

The aged specimens exhibited higher fracture toughness than the RT ones, which is not compatible with the trends observed in all other matrix-dominated test types, including the subsequent Mode II fracture tests. It could be partially related to plasticization effects at the crack front and the deformation mechanism of the faces. However, since the propagation evolves in limited steps, the contribution of each mechanism (strain stored in the faces within a geometrically non-linear state and purely fracture energy) can hardly be identified. For instance, the HW specimens may endure a larger deformation of the faces due to being more ductile as a result of conditioning process, which leads to the determination of higher $G_{Ic}$ values.

### 3.5.4. Static Interlaminar Fracture Toughness Tests Mode II (ASTM D7905)

According to the standard, an initial delamination of 45 mm has been considered by applying a 0.012 mm release film at the middle of the lamination. A head velocity of 0.5 mm/min was applied. A non-precrack (NPC) and a subsequent precrack (PC) loading stage have been applied, each consisting of two tests for compliance calibration and an additional one for crack propagation. In Figure 11, measured load vs. transverse displacement data for RT and HW specimens in the crack propagation test of the reloading stage are presented, whereas measured and predicted instances of crack propagation are also shown. In contrast to the observations made in the DCB test, the woven thermoplastic material exhibits a stable response, similar to the one observed in the case of thermoset UD materials [28]. The FE model succeeds in accurately predicting the slope of the measured curve, whilst it fails to adequately estimate the measured maximum load for crack propagation. This shortcoming of the numerical model is largely attributed to the lack of implementation of appropriate failure criteria for woven thermoplastic materials encompassing interlaminar shear effects.

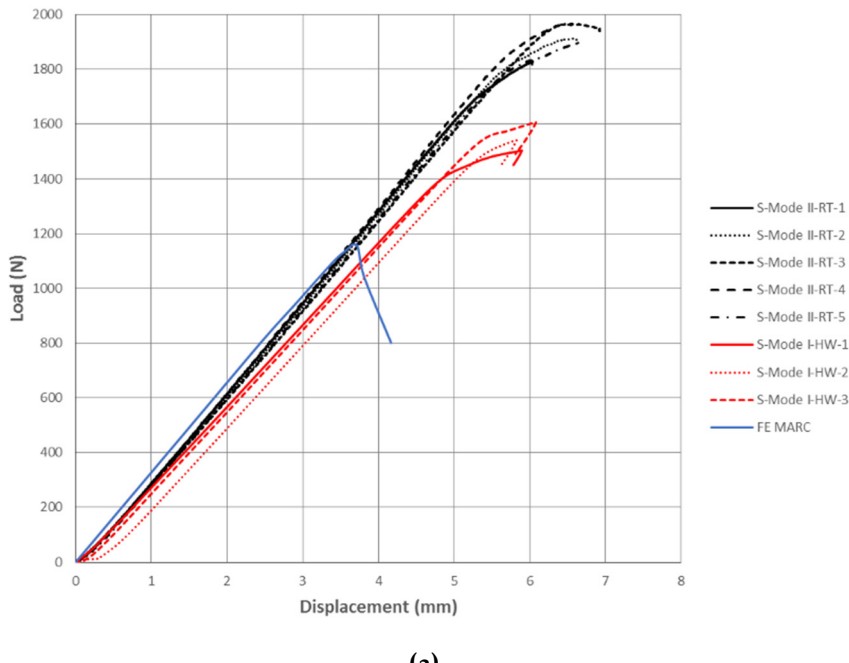

(a)

**Figure 11.** *Cont*.

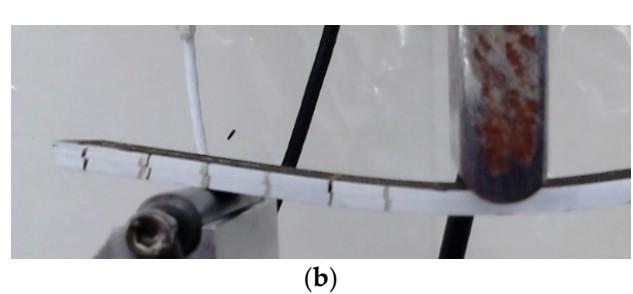
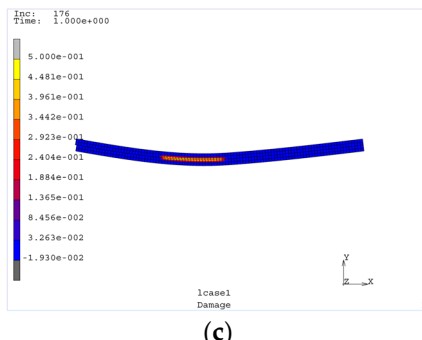

(**b**)                    (**c**)

**Figure 11.** Mode II fracture test: (**a**) measured and predicted load vs. transverse displacement data; (**b**) typical deformed shape during test; (**c**) instance of crack propagation prediction.

Elaboration of the measured data yields the interlaminar fracture toughness values for RT and HW conditions, respectively, presented in Table 11. For this test type, no data are included in the manufacturer datasheet.

**Table 11.** Effect of hot-wet storage aging on interlaminar fracture toughness in Mode II.

| Property | RT | | HW | |
|---|---|---|---|---|
| | **Mean Value** | **CV (%)** | **Mean Value** | **CV (%)** |
| ILFT Mode II($J/m^2$) | 5900 | 8 | 4344 | 7 |

The aged specimens exhibited lower fracture toughness than the RT ones. Since interlaminar fracture toughness is a matrix-dominated mechanical property, that trend is in agreement with the behavior observed in other matrix-dominated tests, such as the IPSS and ILSS tests presented above. Nevertheless, the thickness to width ratio of the specimens involved is in the range of the IPSS ones, which is an additional argument for justifying the higher sensitivity to aging compared to tensile and flexure tests.

3.5.5. Static Interlaminar Fracture Toughness Tests Mode I/II (ASTM D6671)

A mode mixture of $R = G_{II}/(G_I + G_{II}) = 0.2$ has been selected for the RT specimens and a corresponding lever length of $c = 93$ mm. An initial delamination of 45 mm has been considered by applying a 0.012 mm release film at the middle of the lamination. A head velocity of 0.5 mm/min was applied. In Figure 12, measured load vs. opening displacement data for RT specimens are presented, along with instances of measured crack propagation. The "stick-slip" response during crack propagation has been also observed in this case, leading to relatively few data points and large variances. In the case of HW tests, the specimen failed at the root of the upper face at a maximum opening displacement between 3 and 10 mm. Thus, various mode mixtures ($R = 0.25$, 0.3 and 0.5) have been considered, albeit without success in eliminating that trend.

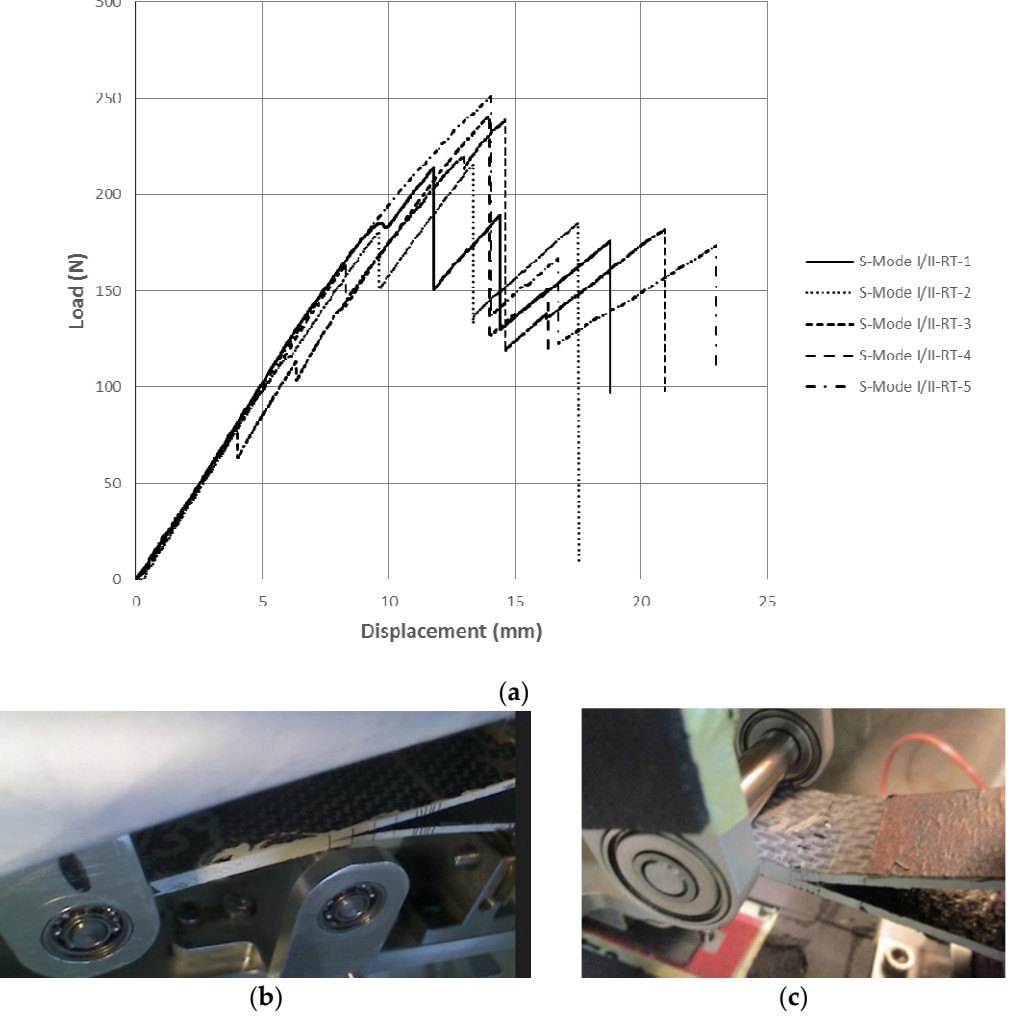

(**a**)

(**b**)                    (**c**)

**Figure 12.** Mode II fracture test: (**a**) measured and predicted load vs opening displacement data RT specimens; (**b**) typical deformed shape during RT test; (**c**) failure mode at root of upper face during HW test.

Elaboration of the measured data yields the interlaminar fracture toughness values presented in Table 12. For this test type no data are included in the manufacturer datasheet.

**Table 12.** Effect of hot-wet storage aging on interlaminar fracture toughness in Mode I/II.

| Property | RT (R = 0.2, 5 sp) | | HW (R = 0.2, 1 sp) | | HW (R = 0.25, 2 sp) | | HW (R = 0.3, 2 sp) | |
|---|---|---|---|---|---|---|---|---|
| | Mean Value | CV (%) | Value | CV (%) | Mean Value | CV (%) | Mean Value | CV (%) |
| ILFT Mode I/II(J/m$^2$) | 3050 | 31 | 521 | - | 814 | 6 | 796 | 14 |

The large variance observed indicates the highly non-linear nature of this test on woven thermoplastic PEEK materials. The aged specimens exhibited significantly lower fracture toughness than the RT ones. However, premature failure of the upper face, as well as the fact that the crack evolves in limited steps, did not allow for a proper estimation of fracture toughness in the case of the HW specimens. The latter two points might indicate that the standard should be further elaborated in order to prescribe a stable and uniform crack growth in woven thermoplastic laminates, especially under hot-wet aged conditions.

## 4. Conclusions

On the basis of the presented study, the following major conclusions may be drawn:

- In all test types, the hot-wet storage aging process led to moisture absorption comparable to values reported in the literature [31];
- Aging was found to significantly affect static properties related to the strength of the thermoplastic woven material in all tests considered, whereas stiffness properties were weakly sensitive to aging. This trend is in line with results reported in the literature for similar types of material [32];
- Matrix-dominated properties, such as interlaminar shear strength and mode II fracture toughness were found to be the most degraded due to aging. A reasonable justification for that, supported by the trends observed in the test campaign, is attributed to the high thickness to width ratio of specimens involved in corresponding tests. This ratio affects the distribution of moisture along the plane of the specimen and becomes dominant beyond the linear elastic part of the response;
- In the case of the DCB and MMB tests, a stick-slip response has been observed, which prevented stable crack propagation and led to relatively large variance in fracture toughness compared to UD thermoset laminates. In these cases, the contribution of each deformation mechanism (strain stored in the faces within a geometrically non-linear state and purely fracture energy) can hardly be identified;
- In the case of the MMB tests on HW specimens, premature failure occurred at the root of the crack upper face, indicating the limited applicability of the standard in such cases;
- Aging was found to have a clearly degrading effect on interlaminar shear fatigue strength. A justification for that trend is the same as in the corresponding static tests;
- Validation of the FE models developed, which were based on effective material properties to be used for initial design purposes, indicated the need to consider failure criteria specifically targeted to thermoplastic woven composite materials in order to accurately capture both load–displacement slope and maximum load in the case of fracture toughness test simulation.

**Author Contributions:** Contribution of each co-author to the current work may be distinguished as: Conceptualization, all; Investigation, K.M. and T.S.P.; Methodology, M.J., K.M., T.S.P., and V.P.; Resources, D.S.-C. and M.M.M.; Validation, M.J., K.M., T.S.P., and V.P.; Visualization, T.S.P.; Writing—original draft preparation, all; Writing—review and editing, T.S.P. All authors have read and agreed to the published version of the manuscript.

**Funding:** The current work has received funding from EU Horizon 2020 Clean Sky II project SHERLOC (Structural Health Monitoring, Manufacturing and Repair Technologies for Life Management of Composite Fuselage) under Grant Agreement No. CS2-AIR-GAM-2014-2015-01.

**Acknowledgments:** The authors from HAI would like to thank our ex-colleague Stavros Kalogeropoulos for his major assistance with the experimental work.

**Conflicts of Interest:** The authors declare no conflict of interest.

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
