# Peer review of "Effect of Hot-Wet Storage Aging on Mechanical Response of a Woven Thermoplastic Composite"

_aerospace, doi:10.3390/aerospace7020018_

Round 1

Reviewer 1 Report

Although there are a lot of test date presented, there is also a lot of information missing: there are hardly no explanations/hypotheses how increases or decreases of test results can be explained; sometimes data are just presented without sufficient text; sometimes choices (e.g. specimen dimensions) are not explained properly.

It would have been better to present less tests but with a sound explanation of the results.

Reviewer 2 Report

The paper entitled "Effect of Hot-Wet Sotrage Aging on Static and Fatigue Response of a Woven Thermoplastic" adresses an interesting topic. A significant amount of experimental work was performed, however, in my point of view, there are too many aspects that I don't find convincing.

1 ) One of the first aspects that I would like to comment is about the specimens dimensions used throughout the paper. It is stated, lines 166 to 167, that "Each specimen had the following nominal dimensions: 250 mm length, 15 mm width and 0.9 mm thickness and a stacking sequence corresponding to [0]3". From what I read in the datasheet from the supplier (https://docplayer.net/87745139-Product-data-sheet-tenax-e-tpcl-peek-hta40.html), the specimens used here are not unidirectionals [0]3 but [0/90]3 composites tested in the warp direction. Therefore, the dimensions are not the ones recommended in ASTM 3039, but should be Length 250 mm, Width 25 mm and thickness 2.5 mm. While it is common and sometimes necessary to differ from the standards, here, I do not understand why these dimensions were chosen. I have the same comment concerning the [90]6 specimens. It is not a [90]6 but a [0/90]6 composite tested in the weft direction. There is no clear reasons why the [0/90]3 and [0/90]6 specimens should have different dimensions. Moreover, the authors could have compared the values from the tests they obtained with the ones from the supplier, which are found within the datasheet. It may also be stated that most of the tests performed in the paper are similar to those found within the datasheet. Again, a comparison would have been interesting.

2) It is stated in lines 53 to 54 that “Thus, it can be concluded that up to date no full characterization of mechanical properties of thermoplastic materials in specimen batch mode has been performed”. This is not untrue, but what is a full mechanical characterization? In the datasheet, mechanical results from more than 20 quasi-static tests are shown, which is in my opinion quite significant. A full mechanical characterization can indeed include fatigue testing, however, less tests were performed in fatigue (3 per condition) than in static (5 per condition). This is rather confusing. The standard states that “a statistically significant distribution of data should be obtained for a given material”.

3) If I understood correctly, the unaged (RT) specimens were tested in one lab and the aged specimens in another lab. While it is common to perform the same tests in different labs (round robin testing), here, comparing specimens in different states in different labs is not common, so it is difficult to trust the results. A comparison between the two labs on the same unaged specimens would have been necessary. Also, when performing an aging study, it is necessary to write in the text how long the specimens were aged in a specific condition and also show the water uptake curves. Non of these two aspects are cited/shown in the text. These aspects were included in your other paper recently published on another material: Plagianakos, T. S., Muñoz, K., Guillamet, G., Prentzias, V., Quintanas-Corominas, A., Jimenez, M., & Karachalios, E. (2019). Assessment of CNT-doping and hot-wet storage aging effects on Mode I, II and I/II interlaminar fracture toughness of a UD Graphite/Epoxy material system. Engineering Fracture Mechanics, 106761.

4) Several pictures of the failure modes are shown and show non valid failures: Fig 2b, failure within the tabs and Fig 12 a and b, failure below the loading noses.

5) Concerning the delamination tests, in mode I, it is stated that the Corrected Beam theory was used. Did you use the correction factor F stated in the standard to account for the large ratio d/a that is higher than 0.4 ? Also, compared with your other paper recently published, less data points are shown in the different delamination figures. This makes it less convincing.

6) The last comment that I have is concerning the y-axis. Why are you plotting the load instead of the stress ? Second, the aim of this paper is to investigate the “Effect of hot wet storage aging”. Therefore, the aim is to compare these two figures if I understand well. Why didn’t you plot all these curves within one Figure? Using one colour for the unaged specimens and another colour for the aged specimens. This may be the simplest and clearest way to highlight the effect of aging. Less figures is always better for a clear understanding. However, if the authors did want two figures, use the same scale on the two figures.

For these reasons, I do not recommend this paper to be published in Aerospace.

Reviewer 3 Report

The authors present a set of results for a thermoplastic material. However, the experimental design is very questionable. They state that the effect of HTW on properties is unique, however, every material used for aerospace must have HTW effect available and most of these results were probably measured before. There is total lack of discussion of the results with the literature. FE models seem to be very simple and not enough described bringing no new information to the reader. Some test such as open hole are not relevant as no comparison is mentioned.

Specific points:

Introduction: references mention only that the properties were studied before however there is not mentioned what results they gained? Their results must be discussed with the presented results at the end of the article or in the results section.

Research design:

1)The authors used test standards, however the sampling size was not fullfilled for fatigue tests meaning the results cannot be considered for comparison. 

3) Open hole results are only for RT, so there is no reason to mention them in the article.

4) Fatigue tests were not performed at 70C so it makes them non relevant in terms of the whole study performed at 70C.

Methods:

1) For bending test, it is not clear whether it is 3 or 4 point bending. There is only 1 span size but failure mode seems to be for 4 point bending.

2) Use tolerance for conditions T = 25C +- ? /45%r.h. +- ?

Results presentation:

1) Remove black borders from graphs

2) The comparisons must be based on hypothesis testing, p-values must be mentioned for all statements. Comments as "rather or seems to be" are not objective.

3) For compression, there is no number of specimens used. Why is the compression strength not evaluated? This should be the most affected property by the HTW conditions and is usually critical for structural design.

Conclusions:

There must be discusion of the results with results in references. There are almost no physical explanations of the presented effects of HTW conditions.

Round 2

Reviewer 1 Report

A lot of work has been put in the responses to the reviewers: compliments for that!

Before publishing I would recommend to add/change some items/address some unclarities:

Explain the stress ratio (0.1) Does the frequency of testing (5Hz) have any effect on the result? Codes like at figure 1 (S-T-O-RT-1) should be explained in the text Explain what CV [%] is what is LMV? Why is the saturation of some test higher than for another (moisture absorption seems orientation independent to me. In plot like figure 6: indicate that the values at 10exp6 are not failing at that point but later by indicating this with an arrow to the right Explain significant differences in values; sometimes it is mentioned as more or less equal Not all additional text are really clarifying. Some are still vague. 

Reviewer 2 Report

The authors have adressed all the comments mentionned in the review, however, I still do not recommend this paper to be published for the following reasons. 

1) While I do believe that some of these results are worth publishing, this paper is clearly a test report rather than a scientific paper. There are far too many figures and not enough discussion. The paper needs to be completely reorganized et resubmitted in my opinion. If I may, let me suggest a reorganization that will help the paper.

Introduction : The idea of this paper is to investigate whether or not hygrothermal aging has an effect on the mechanical properties. Therefore, in my opinion, the text and references you added in the revised version are not relevant. All these references are not about aging. As reviewer 3 stated, carbon/PEEK has been studied for a long time in aviation and aerospace. More references are needed. Moreover, aging has been performed on PEEK itself (non composite) and these types of references are relevant to this study. The introduction needs to be refocused on aging and its effect. Results and discussion : First of all, I would remove the fatigue data. The method and number of specimens tested is criticizable. Second, the water absorption is a section on its own. Depending on the specimens studied, the water content at saturation and the rate of water diffusion need to be compared and discussed. This is essential when performing an aging study. Then, concerning the static data, I would divide them into three main sections : In-plane fibre dominated properties [Tension Warp Weft directions, compression], in-plane matrix dominated properties [+-45] and out-of-plane properties [Four point flexure, ILSS,mode I, II and mixt]. Moreover, concerning the failures, it would be interesting to perform SEM or only optical microscopy to try and explain the differences observed in mechanical properties. Finally I would remove the FE analysis which is not the purpose of this paper.

2) As I stated in the first review, it is very hard to compare results performed within two different labs. A comparison on the unaged specimens is at least needed.

3) Also, while showing the failure modes for all tests reassures the reader, it is not necessary to show them for all tests.

4) Just a comment concerning the merging of unaged data together with HW data within the same figure. Thank you for taking into account the comment, it does help the reader.

Reviewer 3 Report

The authors responded to all comments. Regarding the research design, I still think that the comparisons should be more clear. If you don't want to mention p-values from hypothesis testing, mentioning sample standard deviations instead of CV would better so the reader could see whether the deviation intervals are overlapping or not.
